# Broad activation of latent HIV-1 *in vivo*

Kirston Barton[1], Bonnie Hiener[1], Anni Winckelmann[1,2], Thomas Aagaard Rasmussen[2,3], Wei Shao[4,5], Karen Byth[6,7], Robert Lanfear[8], Ajantha Solomon[3,9], James McMahon[9], Sean Harrington[10,11], Maria Buzon[10,11], Mathias Lichterfeld[10,11], Paul W. Denton[2,12,13], Rikke Olesen[2], Lars Østergaard[2,13], Martin Tolstrup[2,13], Sharon R. Lewin[3,9], Ole Schmeltz Søgaard[2,13] & Sarah Palmer[1]

The 'shock and kill' approach to cure human immunodeficiency virus (HIV) includes transcriptional induction of latent HIV-1 proviruses using latency-reversing agents (LRAs) with targeted immunotherapy to purge infected cells. The administration of LRAs (panobinostat or vorinostat) to HIV-1-infected individuals on antiretroviral therapy induces a significant increase in cell-associated unspliced (CA-US) HIV-1 RNA from $CD4^+$ T cells. However, it is important to discern whether the increases in CA-US HIV-1 RNA are due to limited or broad activation of HIV-1 proviruses. Here we use single-genome sequencing to find that the RNA transcripts observed following LRA administration are genetically diverse, indicating activation of transcription from an extensive range of proviruses. Defective sequences are more frequently found in CA HIV-1 RNA than in HIV-1 DNA, which has implications for developing an accurate measure of HIV-1 reservoir size. Our findings provide insights into the effects of panobinostat and vorinostat as LRAs for latent HIV-1.

[1] Centre for Virus Research, The Westmead Institute for Medical Research, The University of Sydney, 176 Hawkesbury Road, Sydney, New South Wales 2145, Australia. [2] Department of Infectious Diseases, Aarhus University Hospital, Palle Juul Jensens Boulevard 99, Aarhus N 8200, Denmark. [3] The Peter Doherty Institute for Infection and Immunity, The University of Melbourne and Royal Melbourne Hospital, 792 Elizabeth Street, Melbourne, Victoria 3000, Australia. [4] Advanced Biomedical Computing Center, Leidos Biomedical Research Inc., PO Box B, Frederick, Maryland 21702, USA. [5] HIV Dynamics and Replication Program, National Cancer Institute, PO Box B, Frederick, Maryland 21702, USA. [6] NWSLHD Research and Education Network, Darcy Road, Westmead, NSW 2145, Australia. [7] NHMRC Clinical Trials Centre, University of Sydney, NSW 2006, Australia. [8] Department of Biological Sciences, Macquarie University, Sydney, New South Wales 2109, Australia. [9] Department of Infectious Diseases, Alfred Hospital and Monash University, 55 Commercial Road, Melbourne, Victoria 3181, Australia. [10] Ragon Institute of MGH, MIT, Harvard, 400 Technology Square, Cambridge, Massachusetts 02139, USA. [11] Harvard Medical School, 25 Shattuck Street, Boston, Massachusetts 02115, USA. [12] Aarhus Institute of Advanced Studies, Høegh-Guldbergs Gade 6B, Aarhus C 8000, Denmark. [13] Department of Clinical Medicine, Aarhus University, Palle Juul Jensens Boulevard 82 Building B, Aarhus N 8200, Denmark. Correspondence and requests for materials should be addressed to K.B. (email: Kirston.Barton@sydney.edu.au) or to S.P. (email: Sarah.Palmer@sydney.edu.au).

Latent but replication-competent HIV-1 is known to be present in all lineages of circulating T cells[1–4] in the peripheral blood and anatomical compartments[5–9]. Although there appears to be preferential persistence of latently infected cells with distinct integration sites, HIV generally integrates randomly in the host genome[10,11]. To eliminate latently infected cells through activation of transcription, an ideal latency-reversing agent (LRA) must be active within multiple cell types in many tissues, and activate transcription from proviruses integrated into a multiplicity of sites throughout the genome. To date, several independent studies have demonstrated that LRAs activate latent HIV-1 *in vivo* through observations of increased cell-associated unspliced (CA-US) RNA following administration of disulfiram, vorinostat, panobinostat and romidepsin in HIV-1-infected individuals on long-term suppressive antiretroviral therapy (ART)[12–20]. The mechanisms that contribute to HIV latency are diverse and include transcriptional repression due to removal of histone acetylation or methylation, availability of transcription factors, the integration site of the provirus and the availability of the LRA to the infected tissues and cells[21]. Thus, it is conceivable that the use of an LRA that targets a specific mechanism such as histone acetylation may reverse latency in a subset of viruses. Before this study, it was unknown whether the previously observed increases in CA-US RNA were due to activation of a subset of proviruses or to global non-selective activation of a broad spectrum of latent proviruses. Here we show that panobinostat and vorinostat broadly activate HIV proviruses, that a panobinostat-activated virus is genetically similar to that observed during an analytical treatment interruption (ATI) and that the cell-associated RNA contains a large percentage of defective viral sequences.

## Results

**Clinical samples**. To determine whether the histone deacetylase inhibitors (HDACi) panobinostat and vorinostat broadly activate transcription from HIV-1 proviruses *in vivo*, we utilized single-genome/proviral sequencing (SGS/SPS) and phylogenetic analysis with the molecular evolutionary genetics analysis (MEGA) software[22] to compare the genetic composition and diversity of CA RNA and DNA following administration of these compounds during two clinical trials. The vorinostat trial included 20 participants, 15 of whom are included here, who received 400 mg of vorinostat orally once daily for 14 days (Table 1, see original publications for further details[14,23,24]). Significant CA-US RNA increases were observed 8 h after the initial dose of vorinostat and remained elevated at all time points sampled up to 70 days after the last dose of vorinostat[14]. The panobinostat trial included 15 participants who received 20 mg of panobinostat orally three times a week every other week for 8 weeks (Table 1, see original publications for further

details)[14,23,24]. Panobinostat induced significant increases in CA-US RNA at all time points compared with the baseline and increased the detection rate of plasma HIV-1 RNA[23]. Upon completion of panobinostat, nine participants elected to undergo an ATI, during which plasma samples were collected twice weekly to detect rebound virus[14,23,24]. In this study, a total of 2,843 sequences (309 defective and 2,534 intact) from 15 panobinostat trial participants and a total of 1,654 sequences (249 defective and 1,405 intact) from 15 vorinostat trial participants were analysed (Table 1 and Supplementary Tables 1 and 2). Phylogenetic analysis of the sequences from all 30 subjects formed individual clades with no intraparticipant mixing except for sequences from panobinostat participants 1 and 2 who are a known transmission pair (Supplementary Fig. 1). In total, these trials included 26 participants that initiated ART during chronic infection and four participants that initiated ART during acute infection (Table 2). The sequences from the participants who were treated during acute infection had very low diversity between sequences as expected based on previous studies. As a result, the acutely treated participants were excluded from the comparison of CA RNA and DNA sequences here because the sequence diversity is insufficient to detect differences between populations[8,9]. The sequences from the participants who were treated during chronic infection displayed higher levels of diversity as expected because of the prolonged period of viral replication.

**Panobinostat and vorinostat induce broad HIV-1 transcripts**. Peripheral blood samples were collected immediately before, twice during and once following LRA administration. The proviral sequences were obtained to establish a baseline level of diversity to which the induced CA RNA sequences could be compared. To explore the relationship between the proviral DNA and CA RNA sequences obtained during LRA therapy, we first constructed phylogenetic trees for each individual study participant (Figs 1a, 2 and Supplementary Figs 2–28). For the majority of participants, the CA RNA sequences intermingled throughout the phylogenetic tree with the corresponding DNA sequences. Likewise, the genetic diversity, as measured by the mean of the average pairwise distances, of the DNA and CA RNA sequences collected during HDACi dosing was not significantly different between participants (panobinostat 2.5% versus 2.5%, $P > 0.99$; vorinostat 2.9% versus 4.1%, $P = 0.25$; Wilcoxon signed-rank test), indicating that the observed CA RNA arose from a genetically diverse range of proviruses following vorinostat or panobinostat exposure (Figs 1b, 2 and Supplementary Figs 2–28). No difference was observed in the average pairwise distance between the baseline DNA and that collected during panobinostat administration, demonstrating the consistency of the proviral population between these two time points. Furthermore, longitudinal analysis using a linear mixed-effects model of DNA and CA RNA sequences revealed that there was no statistically significant within-participant difference between the baseline and on HDACi DNA sequences (panobinostat mean change 0.02%, s.e. 0.08%, $P = 0.741$; vorinostat mean change 0.13%, s.e. 0.12%, $P = 0.298$; linear mixed-effects model) and no statistically significant within-participant difference between the on HDACi DNA and CA RNA sequences (panobinostat mean difference 0.11%, s.e. 0.33%, $P = 0.748$; vorinostat mean difference 0.35%, s.e. 0.23%, $P = 0.154$; linear mixed-effects model; Supplementary Fig. 29).

In participant 17 from the panobinostat trial, 19 of the 22 RNA sequences from the on-panobinostat time points were found in one clonal expansion that corresponded to a clonal DNA expansion that was detected at all four panobinostat trial time points (Fig. 2). This large clonal expansion was associated with a

**Table 1 | Baseline characteristics of study participants.**

|  | Panobinostat | Vorinostat |
| --- | --- | --- |
| No. of male participants | 15 | 14 |
| No. of female participants | 0 | 1 |
| Age (years)* | 47 (28–53) | 48 (40–56) |
| Virological suppression (years)* | 3.6 (2.5–16.0) | 5.0 (2.7–11.0) |
| Baseline CD4 (cells per μl)* | 935 (615–1,990) | 717 (479–1,136) |
| No. treated during acute infection | 3 | 1 |
| No. of sequences analysed | 2,843 | 1,654 |

CD4, CD4+ T cell.
*Values represent median (range).

**Table 2 | Samples collected and similarity analysis summary for each participant.**

| Participant | Chronic/acute | Pre-ART | ATI | Cell subsets | LPMCs | CD4 DNA‖ATI | CD4 RNA‖ATI | Cell subsets‖ATI | LPMC‖ATI |
|---|---|---|---|---|---|---|---|---|---|
| Pan01 | C | Y | Y | Y | Y | Y | Y | N | N |
| Pan02 | C | Y | Y | Y | N | Y | Y | Y | — |
| Pan04 | A | Y | Y | Y | N | Y* | N | Y* | — |
| Pan05 | A | N | N | N | Y | — | — | — | — |
| Pan06 | C | Y | N | N | N | — | — | — | — |
| Pan07 | C | Y | N | N | Y | — | — | — | — |
| Pan08 | C | N | N | Y | N | N | N | N | — |
| Pan09 | C | N | Y | Y | Y | Y | N | N | Y |
| Pan10 | C | Y | Y | Y | Y | Y | N | N | N |
| Pan12 | C | Y | Y | Y | N | Y | N | Y | — |
| Pan14 | A | Y | N | N | Y | — | — | — | — |
| Pan15 | C | Y | N | N | Y | — | — | — | — |
| Pan17 | C | N | Y | Y | Y | Y | N | N | Y |
| Pan18 | C | Y | Y | N | Y | Y | Y | — | Y |
| Pan19 | C | N | N | N | N | — | — | — | — |
|  | 12/15 | 10 | 9 | 8 | 9 | 8/9 | 3/9 | 3/8 | 3/5 |

A, acute; ART, antiretroviral therapy; ATI, analytical treatment interruption; C, chronic; CD4, CD4+ T cell; LPMC, lamina propria mononuclear cell; N, no, Y, yes.
—, sample not available.
*Acutely treated participant with average pairwise distance less than 0.005.

low average pairwise distance of 0.4%, which is 6.5-fold lower than that of DNA sequences from samples collected during panobinostat administration. This one participant may be a select case in which a single sequence was disproportionally activated by panobinostat, indicating that in some rare cases selective activation does occur. However, because this sequence corresponds to a clonal expansion of DNA sequences, it is not possible to determine whether several cells or a single cell was responsible for the RNA production. In addition, following an HDACi, we observed clonal expansions of identical CA RNA sequences in one of the three vorinostat trial participants and four of the eight panobinostat trial participants who initiated ART during chronic infection and had five or more sequences available for analysis (Supplementary Figs 2–28), which may have occurred because of activation of transcription from identical proviruses in many cells or because of strong activation of a single provirus[25,26]. Importantly, each tree that contained a cluster of identical sequences also contained several unique CA RNA sequences that were evenly distributed throughout the DNA sequences, further confirming that vorinostat and panobinostat activate transcription from a broad range of proviruses.

**A large fraction of cell-associated HIV-1 RNA is defective.** HIV-1 sequences have up to 8% genetic diversity within a single infected individual[27]. This is largely due to the low-replication fidelity of reverse transcriptase and to hypermutation by APOBEC3G (refs 28, 29). We quantified the percentage of defective sequences (that is, containing stop codons or deleterious hypermutation) in both the DNA and CA RNA from the two samples during HDACi dosing. We found that the CA RNA had a significantly higher percentage of defective sequences than the DNA in CD4+ T cells from participants who received panobinostat and vorinostat (panobinostat 29.11% RNA versus 10.2% DNA, $P = 0.008$; vorinostat 45.5% RNA versus 11.36% DNA, $P = 0.03$; Wilcoxon signed-rank test; Fig. 3). This same relationship was observed in samples collected before and following LRA administration, indicating that vorinostat and panobinostat do not selectively activate transcription from hypermutated sequences but rather uniformly increase transcription from all proviruses. Furthermore, longitudinal analyses using a linear mixed-effects model also revealed that, after adjusting for time, a significantly higher percentage of CA

RNA sequences were defective compared with DNA sequences for participants receiving panobinostat and vorinostat (panobinostat 3.6-fold: 95% confidence interval (CI) 2.0–6.3, $P < 0.001$; vorinostat 2.8-fold: 95% CI 1.3–5.9, $P = 0.019$; linear mixed-effects model; Supplementary Fig. 30 and Supplementary Tables 3 and 4). In comparison, only 5 of the 246 pre-ART plasma RNA sequences and 8 of 183 post-ATI RNA plasma samples were found to be defective in the panobinostat trial. These findings indicate that, as expected, defective virus does not significantly contribute to plasma RNA before initiation of ART or following ART discontinuation.

**ATI viraemia is seeded by the peripheral blood and intestine.** Next, we compared DNA sequences collected during panobinostat administration to sequences from the plasma HIV-1 RNA that emerged during the ATI. We obtained a total of 183 plasma ATI sequences from the nine ATI participants. In contrast to the diverse pattern of CA RNA sequences, each of the nine phylogenetic trees contained from one to nine clusters of identical sequences from the ATI (Fig. 2). Furthermore, the average pairwise distance of the plasma HIV-1 RNA sequences during ATI was significantly less than that of the DNA sequences from samples collected while receiving panobinostat (Fig. 1b, 1.5% ATI to 2.5% DNA, $P = 0.031$; Wilcoxon signed-rank test). Longitudinal analyses that were performed using a linear mixed-effects model also revealed a statistically significant within participant decrease in genetic diversity of the RNA sequences that were collected during the ATI compared to the DNA sequences collected while the participants were on panobinostat (mean change $-0.72\%$, s.e. 0.55%, $P = 0.043$; linear mixed-effects model). This reduction in viral diversity supports the observation from the phylogenetic trees that the initial rebound virus emerged from a small subset of existing proviral sequences as observed previously[30–35].

In total, we identified HIV-1 DNA that was identical to plasma RNA from the ATI in eight out of the nine total participants. In addition, when we compare those treated during chronic infection who elected to undergo the ATI and from whom we recovered more than five sequences, in four out of seven participants, we identified DNA sequences from samples collected while taking panobinostat that were identical to expansions of HIV-1 RNA detected in plasma during the ATI (Fig. 2; Pan09, Pan10, Pan17 and Pan18). Notably, three participants had ATI sequences that were similar to CA RNA sequences collected while the participant

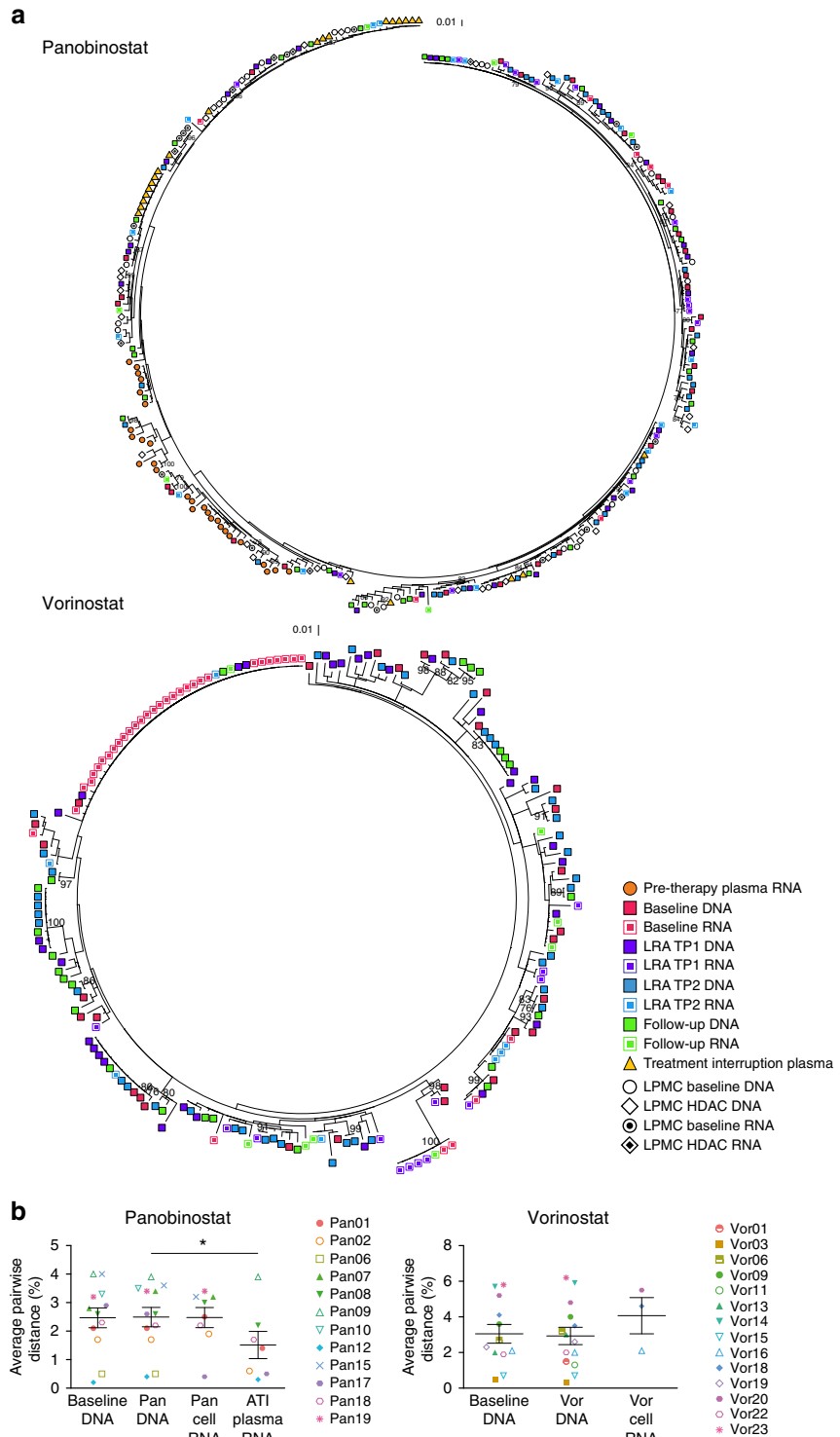

**Figure 1 | Panobinostat and vorinostat non-selectively activate transcription from latent HIV-1 proviruses.** (**a**) Representative phylogenetic trees of HIV-1 sequences from HIV-infected participants on suppressive ART who received panobinostat (Pan18) or vorinostat (Vor16) showing the genetic relationship of sequences from each time point. For participant Pan18, the plasma samples were collected ~1 year and 6 months before initiation of antiretroviral therapy and 14 days following the analytical treatment interruption. Peripheral blood samples were collected at baseline, 2 h after the first dose of panobinostat (TP1), 32 days after the first dose of panobinostat (TP2) and 38 days after the final panobinostat dose. Intestinal lamina propria mononuclear cells were collected at baseline (1 week before the first panobinostat dose) and during week 4 of the panobinostat trial. For participant Vor16, peripheral blood samples were collected at baseline, 7 days after the first dose of vorinostat (TP1), 14 days after the first dose of vorinostat (TP2) and 7 days after the final vorinostat dose. (**b**) Average pairwise distance of cell-associated DNA (Pan $n = 12$, Vor $n = 12$) before and DNA (Pan $n = 12$, Vor $n = 14$) and cell-associated RNA (Pan $n = 8$, Vor $n = 3$) during vorinostat and panobinostat administration, as well as the plasma HIV-1 RNA following an ATI for the panobinostat trial ($n = 7$). Each data point represents the group mean ± s.e.m. The Wilcoxon signed rank test was used to generate the $P$ values. *$P \leq 0.05$.

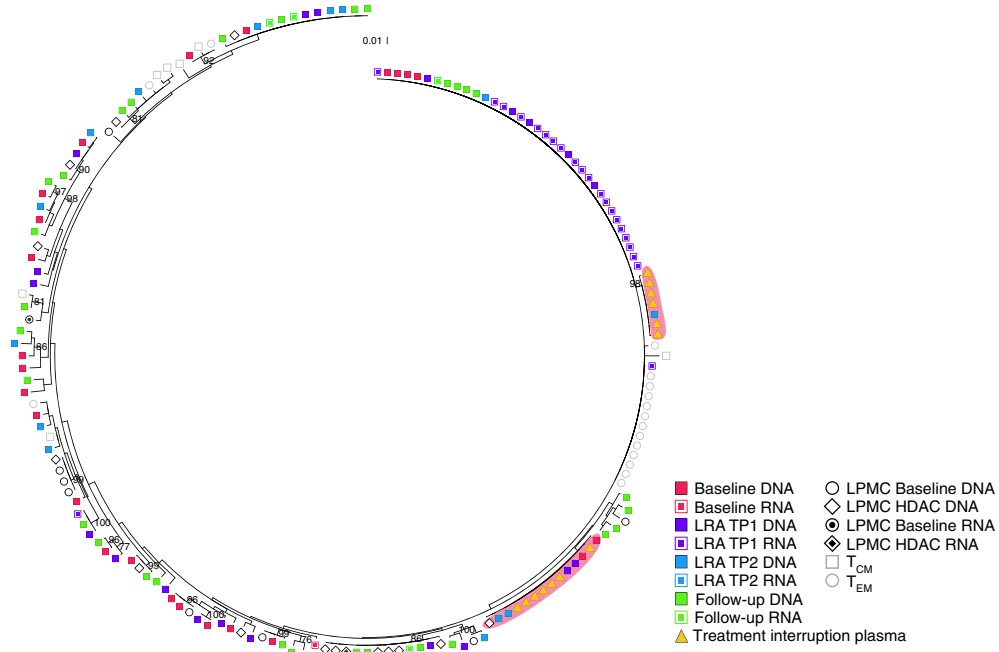

**Baseline DNA** (filled red square)
**Baseline RNA** (red outlined square)
**LRA TP1 DNA** (filled purple square)
**LRA TP1 RNA** (purple outlined square)
**LRA TP2 DNA** (filled blue square)
**LRA TP2 RNA** (blue outlined square)
**Follow-up DNA** (filled green square)
**Follow-up RNA** (green outlined square)
**Treatment interruption plasma** (orange triangle)
○ **LPMC Baseline DNA**
◇ **LPMC HDAC DNA**
◉ **LPMC Baseline RNA**
◈ **LPMC HDAC RNA**
□ **T_CM**
○ **T_EM**

**Figure 2 | Proviruses in CD4$^{+}$ T cells contribute to plasma viraemia during an analytical treatment interruption.** Representative phylogenetic tree of HIV-1 sequences (from participant 17) showing that two clusters of identical HIV-1 sequences were detected in the plasma collected during the ATI that are identical to clonal expansions of cell-associated DNA sequences collected during panobinostat administration (red background). The plasma samples were collected 10 and 14 days following the analytical treatment interruption. Peripheral blood samples were collected at baseline, 1 day after the first dose of panobinostat (TP1), 18 days after the first dose of panobinostat (TP2) and 38 days after the final panobinostat dose. Intestinal lamina propria mononuclear cells were collected at baseline (1 week before the first panobinostat dose) and during week 4 of the panobinostat trial. Cell subsets were sorted from peripheral blood samples collected 38 days after the final panobinostat dose.

was receiving panobinostat on ART (Fig. 2; Pan01, Pan02 and Pan18; > 99% similarity), indicating that oral panobinostat effectively targeted proviruses that have the ability to contribute to plasma viraemia during treatment interruption (Table 2).

Five individuals who participated in the ATI also provided intestinal lamina propria mononuclear cells (LPMCs) at baseline and during the final week of panobinostat dosing. In one participant (Pan17), a DNA sequence from the LPMCs was identical to plasma ATI sequences and in three participants (Pan09, Pan17 and Pan18) DNA from the LPMCs was highly similar to plasma ATI sequences (> 99% similarity). In addition, in one participant we observed CA RNA that had > 99% sequence similarity to sequences identified in plasma during the ATI (Pan18). These results indicate that reservoir virus in LPMCs in the intestine is capable of contributing to viraemia following ATI and is the first *in vivo* observation of a tissue-derived cell that carried provirus-matching rebound plasma virus.

**Virus able to rebound persists in clonally expanded cells.** Previous studies have proposed that clonally expanded cells only contain defective proviruses[10]. Therefore, we examined the sequences in this study to determine whether this hypothesis was universally true. In participant 17, we observed sequences from plasma HIV-1 RNA during ATI that were identical to an expanded population of DNA sequences. In agreement with Kearney *et al.*[36], this observation indicates that proviruses in cells that have undergone clonal expansion are able to contribute to rebound following ATI. While uncommon, this example demonstrates that HIV-1 proviruses that are able to contribute to viraemia can persist in proliferating cells. These findings are in contrast to those of Chun *et al.*[31], who found that HIV-1 in the

plasma following treatment interruption was distinct from that detected in the latent reservoir in resting CD4$^{+}$ T cells[10].

**Central memory cells contribute to rebound viraemia.** We also sorted CD4$^{+}$ memory T-cell subsets including Naive, stem cell (T_SCM), central memory (T_CM), effector memory (T_EM) and terminally differentiated (T_TD) cells from eight participants. These samples were collected 38 days after the final dose of panobinostat. In total, 251 HIV-1 DNA sequences were obtained from the various subsets from eight participants (35 Naive, 20 T_SCM, 77 T_CM, 102 T_EM and 17 T_TD). The percentage of defective sequences available from each subset was similar to that observed for the overall CA DNA discussed above (2.9% Naive, 5% T_SCM, 7.8% T_CM, 10.4% T_EM and 5.9% T_TD). All eight participants from whom we had cell subsets also participated in the ATI. In two participants treated during chronic infection, we identified T_CM sequences that were closely related to sequences from the treatment interruption (Table 2; Pan02 and Pan12; similarity > 99.8%). However, none of the other subsets contained sequences that were closely related. These data do not definitively indicate that the treatment interruption viraemia does not come from these subsets, but rather is likely a result of the limited sample size available for analysis.

SGS/SPS has some limitations in this context. While we were able to detect defective viruses, it is possible that the intact sequences are part of a larger defective sequence. Our PBMC samples were collected from the upper extremity, which may lead to some sampling bias. However, the sequences collected from each time point and sample intermingled throughout the phylogenetic tree indicating that the sampling bias was minimal. As with all limiting dilution assays, some minor viral variants may not be detectable at the lower dilutions. We limited our

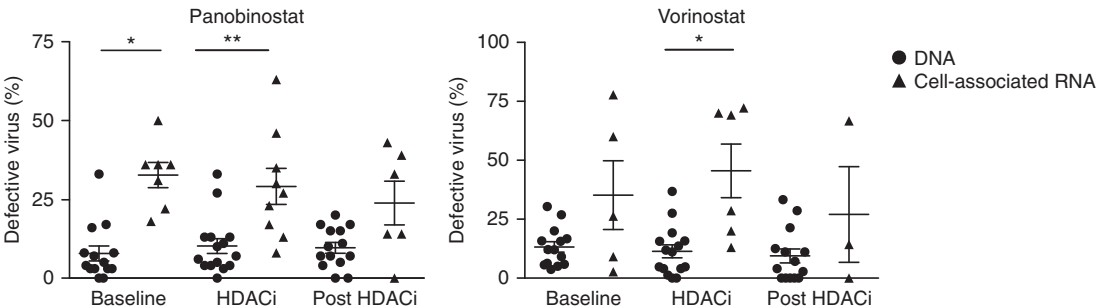

**Figure 3 | A significant proportion of cell-associated HIV-1 RNA is defective.** The percentage of defective virus detected in CD4[+] T cells collected from HIV-infected participants on ART before, during and following administration of panobinostat (baseline DNA $n = 14$, RNA $n = 7$; on HDACi DNA $n = 15$, RNA $n = 9$; post HDACi DNA $n = 14$, RNA $n = 6$) or vorinostat (baseline DNA $n = 14$, RNA $n = 5$; on HDACi DNA $n = 15$, RNA $n = 6$; post HDACi DNA $n = 14$, RNA $n = 3$). Each data point represents the group mean, and the error bars represent the s.e.m. *$P \leq 0.05$, **$P \leq 0.01$. The Wilcoxon signed rank test was used to generate the $P$ values.

analyses of the average pairwise distance to samples from which we obtained more than five intact sequences to reduce bias. We performed a simulation in which we used the cell-associated RNA and DNA sequences that were collected at baseline, during HDACi administration or during follow-up as a model population ($n = 219$, average pairwise distance = 3.33%). The model population was subsampled for 2–50 sequences, with 1,000 replicate subsamples at each sequence number. Five sequences were determined to be the lowest number of sequences with an acceptable level of confidence in the calculated average pairwise distance (95% CI 3.281–3.410, Supplementary Fig. 31).

## Discussion

In conclusion, we detected HIV-1 DNA from peripheral blood CD4[+] T cells that was identical to plasma HIV-1 RNA that emerged during an ATI. These data demonstrate that the proviruses that contribute to plasma viraemia following ART cessation can be identified in CD4[+] T cells. Importantly, panobinostat activated transcription from proviruses in CD4[+] T cells and LPMCs that were genetically similar to those observed in the plasma during an ATI, indicating that it activated transcription from virus that contributed to viral rebound following ART discontinuation. Specifically, central memory cells were demonstrated to contain HIV DNA that was similar to rebound viraemia, which highlights the important role of this subset in persistence of latent HIV-1. Furthermore, we demonstrated that a significant percentage of the detected CA RNA sequences contained stop codons and/or were hypermutated, making them defective. The high percentage of hypermutated CA RNA that was detected emphasizes the importance of developing cost-effective sensitive assays that measure replication-competent virus when assessing the activity of LRAs. Finally, our study demonstrates that panobinostat and vorinostat broadly activated transcription from genetically diverse HIV-1 proviruses *in vivo*, which is promising for the development of future HDACi-based therapies that aim to activate latent HIV-1 proviruses as part of an eradication strategy.

## Methods

**Nucleic acid extraction and cDNA synthesis.** For the vorinostat trial, we obtained half of the DNA and RNA extracted from one million CD4[+] T cells from each participant before the initial vorinostat dose, at the two time points with the highest CA-US RNA measurement during vorinostat dosing (varied from 2 h to 14 days post-initial vorinostat dose) and 7 days after the final dose of vorinostat. For the panobinostat trial, we obtained one million CD4[+] T cells before the first dose of panobinostat, at the two time points with the highest CA-US RNA measurement (which varied from 2 h to 46 days after panobinostat administration for each participant) and 38 days after the final panobinostat dose. An additional on-HDAC therapy sample was analysed for panobinostat participant 2 time point 2 on day 42

post-panobinostat administration because of the low number of sequences obtained from the initial time point 2 sample.

For the samples from the vorinostat trial, CD4[+] T cells were isolated from 10 million PBMCs using the CD4[+] T-cell isolation kit from Miltenyi Biotec (Cat. no. 130-096-533). At least one million CD4[+] T cells were obtained for each participant. For the panobinostat trial, one million CD4[+] T cells were used to extract the RNA and DNA, and we received half of the total product for each. The cells were lysed using QIAshredder (Qiagen, Cat. no. 79565), and then, CA RNA and DNA were extracted from the cell lysate with the AllPrep DNA/RNA Mini Kit (Qiagen, Cat. no. 80204) according to the manufacturer's instructions. For the panobinostat trial, nucleic acids were isolated from the LPMCs using the Allprep Isolation Kit (Qiagen #80204)[24].

Plasma was collected before the initiation of antiretroviral therapy and following an ATI. Plasma was ultracentrifuged and extracted using a guanidinium-based method[37,38].

Plasma HIV-1 RNA and CA RNA were reverse-transcribed to cDNA using the Superscript III (Life Technologies, Cat. No. 11752250) cDNA Synthesis Kit and a gene-specific primer (E115 Reverse: 5′-AGAAAAATTCCCCTCCACAATTAA-3′) according to the manufacturer's instructions.

**Single-genome sequencing.** To characterize the HIV-1 genetic populations before the initiation of antiretroviral therapy, before, during and after HDAC therapy, and after an ATI, we performed SGS/SPS[8,9,27,37,39]. cDNA and/or the HIV-1 DNA extracted from cells were serially diluted to a single copy. The V1–V3 region of HIV-1 *env* was amplified from the DNA or cDNA by two rounds of nested PCR amplification. The following primers were used for amplification: Round 1 forward (E20) 5′-GGGCCACACATGCCTGTGTACCCACAG-3′ and reverse (E115) 5′-AGAAAAATTCCCCTCCACAATTAA-3′; round 2, forward (E30) 5′-GTGTACCCACAGACCCCAGCCCACAAG-3′ and reverse (E125) 5′-CAATTTCTGGGTCCCCTCCTGAGG-3′. For round 1 of PCR, the following thermocycler parameters were used: 94 °C for 2 min, 94 °C for 30 s, 52 °C for 30 s, 72 °C for 1 min, 44 cycles of steps 2–4 and 72 °C for 3 min. For round 2 of PCR, the following thermocycler parameters were used: 94 °C for 30 s, 55 °C for 30 s, 72 °C for 1 min, 41 cycles of steps 1–3 and 72 °C for 3 min. The PCR products representing single HIV-1 sequences were sequenced using Sanger sequencing (Australian Genome Research Facility, Sydney, Australia).

**Phylogenetic analysis.** Contigs were generated from the raw sequencing data using an in-house computer programme written in Perl scripting language (available upon request). Vigorous automated and manual quality-control parameters were used to eliminate low-quality sequences before and following the generation of the contigs. Multiple alignment files were created for each participant using MUSCLE[40]. Defective virus was characterized using the Los Alamos HIV Database Hypermut tool (http://www.hiv.lanl.gov) to screen for HIV-1 DNA and RNA sequences containing G-A hypermutations and by visually screening the amino-acid sequences for premature stop codons. Defective viruses were excluded, and the remaining sequences were used to construct maximum likelihood phylogenetic trees using MEGA-CC[22]. An appropriate model for nucleotide substitution was determined for each phylogenetic tree using MEGA-CC model finder. The models used to generate the phylogenetic trees in this study included the following: generalized time-reversible model, Tamura 3-parameter model, Hasegawa-Kishino-Yano model and Tamura Nei model incorporating gamma-distributed (gamma category 4) and/or invariant sites where appropriate. Our heuristic tree search strategy used the nearest neighbour interchanges branch-swapping algorithm. Branch support was inferred using 1,000 bootstrap replicates. Measurements of genetic HIV-1 diversity (average pairwise distance) of HIV-1 DNA or RNA sequences were calculated using the p-distance

model in MEGA-CC. Branches with less than one bootstrap support were removed using TreeCollapseCL4 (Emma Hodcroft, http://emmahodcroft.com/TreeCollapseCL.html). Bootstrap values greater than 75 are included on the phylogenetic trees. Tree images were generated with ggtree (https://bioconductor.org/packages/release/bioc/html/ggtree.html). Participants treated during acute infection with an average pairwise distance of less than 0.005 were excluded from the comparison of CA RNA with DNA because the diversity in the sample was insufficient to detect differences between the populations (Vorinostat participant Vor02 and Panobinostat participants Pan04, Pan05 and Pan14). Samples with fewer than five sequences were excluded from analyses because of insufficient data. The phylogenetic trees for all participants can be found in Supplementary Figs 2–28.

**Statistical analyses.** Statistical analysis for the interparticipant analysis using the Wilcoxon test was performed with Prism 6 for Mac OS X. All values in the graphs are expressed as the mean ± s.e.m. Participants who initiated ART during acute infection were excluded from the interpatient analyses of the average pairwise distance. Only samples with five or more intact sequences were included in the comparisons of the average pairwise distance and samples with five or more total sequences were included in the hypermutation analyses. A Gaussian distribution could not be established for all groups because of sample size. Therefore, statistical comparisons were made using Wilcoxon test, which does not assume Gaussian distribution. A paired $t$-test was also performed for each comparison, and the significance results for the Wilcoxon and paired $t$-test were the same for each. The $P$ values reported are from the more conservative Wilcoxon test. No differences in the variance between the compared groups were detected. A $P$ value less than 0.05 was considered significant.

For the additional linear mixed-effect model statistics shown in the Supplementary Information, the statistical software S-PLUS 8.2 was used to analyse the data. Two-tailed tests with a significance level of 5% were used throughout. Patient identifier was considered as a random effect and the sample type factor as both a fixed effect and as a random effect with a general positive definite covariance structure. The percentage of dead-end virus was log-transformed to approximate normality and to stabilize the variance before analysis. Linear mixed-effects models were used to investigate the joint effects of time (treated as a three-level factor) and type (DNA or RNA) on log (percentage of dead-end virus) within each drug trial. Patient identifier, time and type were considered as random effects with a general positive definite covariance structure, and time, type and their two-way interaction as fixed effects. Parameter estimates and their 95% CIs for the log (percentage of dead-end virus) analyses were back-transformed to present results using the original scale of measurement. Diagnostic plots were used to assess the adequacy of the fitted models. For each fitted model, these included scatterplots of standardized residuals by fitted values and observed versus fitted values. Normal quantile plots (Q–Q plots) of residuals and of estimated random effects were used to check the assumption of normality for the within-patient errors and for the random effects.

**Simulation of sampling.** The cell-associated DNA and RNA sequences from the HDAC trial time points were used to generate a matrix of average pairwise distances (average pairwise distance 3.33 from 219 sequences). To calculate the bias and variance associated with estimating the average pairwise distance from smaller numbers of sequences, the matrix was subsampled for 2–50 sequences 1,000 times each, and the resulting pairwise distance estimates were examined graphically (Supplementary Fig. 31). The packages ggplot and plyr in R language and environment were used to perform the simulation and plot the resulting data[41].

**Code availability.** The in-house computer programme used to generate contigs of sequences from the raw sequencing data is available from the corresponding author on request with no restrictions.

**Data availability.** All sequences from the study have been deposited in the GenBank Nucleotide database with the accession codes KU563159 to KU563207, KU609626 to KU612115 and KU660076 to KU661479. The rest of the data that support the conclusions of this study are available from the corresponding authors upon request.

**Ethics approval.** Informed consent was obtained from all study participants. The vorinostat trial was approved by the Alfred Human Research Ethics Committee, and the study is registered at ClinicalTrials.gov (NCT01365065). The panobinostat trial was approved by the Danish scientific ethical committee for the central Jutland region in accordance with the principles of the Declaration of Helsinki.

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

## Acknowledgements

This work was supported by the Delaney AIDS Research Enterprise (DARE) to Find a Cure 1U19AI096109, the National Institutes of Health grant 1R21AI113096 (S.P., S.R.L. and T.A.R.), amfAR grant 108830 with support from FAIR: the Foundation for AIDS and Immune Research, the amfAR Research Consortium on HIV Eradication (108928-56-RGRL) and the National Health and Medical Research Council (NHMRC) of Australia (grant APP1061681). S.R.L. is supported by a practitioner fellowship and J.M. by an early career fellowship from the NHMRC of Australia. T.A.R., L.Ø., M.T., O.S.S. and S.R.L. were supported by a grant from the Danish Council for Strategic Research (grant #0603-00521B). We thank Dr Timothy Schlub for statistics consultation. We acknowledge the participation and commitment of study participants, which made the study possible.

## Author contributions

K. Barton designed and performed experiments, analysed data and wrote the manuscript. B.H. designed and performed experiments, analysed data and wrote the Methods section. A.W. designed and performed experiments and analysed data. W.S. prepared sequences for analysis. K. Byth performed the mixed-effect model statistical analyses. R.L. performed the simulation of sampling analyses. A.S., P.W.D., S.H., M.B., M.L. and R.O. were involved in sample processing and preparation and assisted with manuscript preparation. J.M. provided samples and assisted with manuscript preparation. T.A.R., M.T., L.O., O.S.S. and S.R.L. designed the study, provided samples and assisted with manuscript preparation. S.P. designed the study, supervised the work performed and edited the manuscript.

## Additional information

**Competing financial interests:** S.R.L. and J.M. received funding from Merck for the investigator-initiated clinical trial of vorinostat described in this manuscript. Payment was made to their institution. The remaining authors declare no competing financial interests.

