## [Peer Review File · Nature Communications]

Reviewers' comments:

Reviewer #1 (Remarks to the Author):

A. Here the authors use single genome sequencing to assess the level of viral diversity following latency reversing therapy as a surrogate for activation of many viral cells and cell types. These data support the hypothesis that administration of panobinostat and vorinostat activates a broad range of cells.

B. This work is original and of considerable interest to the field.

C. The data and methodology are cutting edge for sequencing.

D. This modified version has more detailed and meaningful statistics.

E. Conclusions are accurate and are supported.

F. No suggested improvements.

G. References are appropriate.

H. The manuscript is well written and clear.

Note from Editor: Referee #1 is satisfied with your responses to the points previously raised by referee #2, who was not available for review.

Reviewer #3 (Remarks to the Author):

The authors have answered the technical issues raised by all the reviewers in a satisfactory manner. The impact of the study is still somewhat an issue since it is not surprising that broad activators activate a heterogeneous population of viruses. Nonetheless it is worthwhile to document this rigorously.

Author's response:

Thank you to the reviewers for your careful and detailed review of this manuscript. Your comments and suggestions were highly appreciated and resulted in a stronger manuscript. We look forward to publication and dissemination of our work to the broader scientific community and hope that it will inform future work in the field of persistent HIV.